



# Estimation of twenty-four-hour continuous cloud cover using ground-based imager with convolutional neural network

Bu-Yo Kim, Joo Wan Cha, Yong Hee Lee

Research Applications Department, National Institute of Meteorological Sciences, Seogwipo, Jeju, 63568, South Korea

*Correspondence to*: Bu-Yo Kim (kimbuyo@korea.kr)

**Abstract.** In this study, we aimed to estimate cloud cover with high accuracy using images from a camera-based imager and a convolutional neural network (CNN) as a potential alternative to human-eye observation on the ground. Image data collected at 1 h intervals from 2019 to 2020 at a manned weather station, where human-eye observations were performed, were used as input data. The 2019 dataset was used for training and validating the CNN model, whereas the 2020 dataset was used for

testing the estimated cloud cover. Additionally, we compared satellite (SAT) and ceilometer (CEI) cloud cover to determine the method most suitable for cloud cover estimation at the ground level. The CNN model was optimized using a deep layer and detailed hyperparameter settings. Consequently, the model achieved an accuracy, bias, root mean square error (RMSE), and correlation coefficient (R) of 0.92, –0.13, 1.40 tenths, and 0.95, respectively, on the test dataset, and exhibited approximately 93% high agreement at a difference within ±2 tenths of the observed cloud cover. This result demonstrates an

improvement over previous studies that used threshold, machine learning, and deep learning methods. In addition, compared with the SAT (with an accuracy, bias, RMSE, R, and agreement of 0.89, 0.33 tenths, 2.31 tenths, 0.87, and 83%, respectively) and CEI (with an accuracy, bias, RMSE, R, agreement of 0.86, –1.58 tenths, 3.34 tenths, 0.76, and 74%, respectively), the camera-based imager with the CNN was found to be the most suitable method to replace ground cloud cover observation by humans.

## 1 Introduction

In general, clouds are a well-known natural phenomenon that plays an important role in balancing atmospheric radiation and global heat, the hydrological cycle, and weather and climate changes in the atmosphere–Earth system (Shi et al., 2021; Zhou et al., 2022). Ground cloud cover observation data are particularly important for weather prediction, environmental monitoring, and climate change prediction (Krinitskiy et al., 2021). In addition, cloud cover is an important meteorological factor for solar

energy-related research fields; aviation operation-related businesses; and monsoon, El Niño, and La Niña studies based on ocean observations (Taravat et al., 2014). Ground cloud cover estimation is currently being automated with instrumental observation; however, thus far, it has been combined with human-eye observation according to the synoptic observation guidelines of the World Meteorological Organization (WMO, 2021). Human-eye observation may lack consistency depending on the observer's condition or observation period and is performed at a low frequency temporally (Kim et al., 2021b). Therefore,

automatic observations are essential to reduce the uncertainty in cloud observations and increase periodicity. However, clouds



have different optical properties according to their shape, type, thickness, and height (Wang et al., 2020); thus, instrument-based cloud detection and cloud cover estimation on the ground remain challenging.

Ground-based methods for remote and automatic observation and estimation of cloud cover can be divided into meteorological satellites (SATs), ceilometers (CEIs), and camera-based imagers. Meteorological SATs have the advantage of obtaining

observational data over a wide spatial range and at dense temporal intervals. However, the uncertainty of cloud detection (Kim et al., 2018; Zhang et al., 2018) and bias due to geometric distortion are large, depending on the height of the cloud (Sunila et al., 2021). In the case of a CEI, a narrow-width beam is emitted vertically, clouds are detected from the returned signal strength, and cloud cover is estimated by weighting the previously detected cloud information (Kim et al., 2021b). Therefore, an incorrectly estimated cloud cover can be obtained even if the cloud does not fall within the beam width range of the CEI or if

it is not detected. Nevertheless, it is used at several weather stations owing to the advantage of "automatic observation." In the case of camera-based imagers, cloud cover can be estimated using the color information in an image captured from a sky-dome image that is similar to human-eye observations (Kim and Cha, 2020; Kim et al., 2016). Imagers are widely used to estimate cloud cover estimation, and the estimation accuracy is higher than that of other remote observations (Alonso-Montesinos, 2020; Kim and Cha, 2020).

Cloud cover estimation using ground-based imagers can be performed using traditional methods, machine learning (ML), and deep learning (DL). Traditional methods estimate the cloud cover by setting a constant (or variable) value for the ratio or difference in red, green, and blue (RGB) color in the image as a threshold and distinguishing between cloud and sky (non-cloud) pixels (Shi et al., 2021; Wang et al., 2020). However, empirical methods do not adequately distinguish between the sky and clouds under various atmospheric and light source conditions (Kim and Cha, 2020; Kim et al., 2015; Kim et al., 2016;

Yang et al., 2015). By contrast, ML and DL methods have achieved relatively highly accurate cloud cover estimation results, addressing the limitations of empirical methods through image learning (Kim et al., 2021b; Xie et al., 2020). In particular, DL methods can learn images deeper than ML methods; therefore, they can hierarchically extract various contextual information and global features from images to statistically estimate the cloud cover (Wang et al., 2020). Various algorithms are used for image learning, with the convolutional neural network (CNN) being the most commonly used. A CNN can design a model

with high accuracy by setting the hyperparameters, such as the layer depth, feature size, and activation function, according to the characteristics of the input data. The aim of this study was to estimate cloud cover with high accuracy using images from a camera-based imager and a CNN as a potential alternative to human-eye observation on the ground. The estimated cloud cover was evaluated by comparing cloud cover data observed from manned weather stations, meteorological SATs, and CEIs.

## 2 Research data and methods

### 2.1 Imager information and input dataset

The camera-based imager used in this study was an automatic cloud observation system (ACOS) developed by the Korea Meteorological Administration (KMA), National Institute of Meteorological Sciences (NIMS), and A&D System Co., Ltd., as



shown in Fig. 1. The ACOS is installed in the Daejeon Regional Office of Meteorology (36.37°N, 127.37°E) and continuously captures the sky-dome over a 180° field-of-view (FOV) (fisheye lens) for 24 h at 10 min intervals and saves it as images. As

summarized in Table 1, the ISO and exposure time were automatically set such that the objects (clouds) could be continuously captured during the day and night (Kim et al., 2021b). In addition, a heating and ventilation device was installed such that the clouds can be captured without artificially managing the ACOS. The image data used were captured at 1 h intervals from January 2, 2019, to December 9, 2020, considering the human-eye observation interval. Instances of equipment maintenance, power outages, or unfavorable weather conditions, such as snow cover or fog preventing the capture of images or making it

impossible to visually identify cloud cover, were excluded from the image data. In addition, KMA cloud cover observations were performed at 1 h intervals during the day and at 1–3 h intervals at night (Kim and Cha, 2020). Therefore, 5,607 and 4,742 images were collected in 2019 and 2020, respectively, excluding the unavoidable cases.

In this study, the 2019 dataset was used for training and validating the CNN model, whereas the 2020 dataset was used for testing. For training and validating the CNN model, 500 data points were randomly sampled with a replacement for each cloud

cover class in 2019, and 5,500 data points were randomly selected at a ratio of 8:2 to form training (4,400 cases, including duplicate data) and validation (897 cases, excluding duplicate data) datasets. The frequency distribution by class of observed cloud cover data in 2019 was as high as approximately 61% in the 0-tenth class (approximately 36%) and 10-tenths class (approximately 26%), and the remaining 1–9 tenths class exhibited a low-frequency distribution of approximately 39% (minimum at 1 tenth class: approximately 2%, maximum at 9 tenths class: approximately 8%). Therefore, when training all

the data from 2019, the model can overfit for the 0-tenth and 10-tenths classes. To prevent the potential overfitting that can occur in specific classes, we employed random sampling with replacement, which limits the number of images for each cloud cover class. This approach ensures that the model is designed and trained such that there is sufficient weight update in each layer of the CNN for classes with fewer cases (Park et al., 2022).





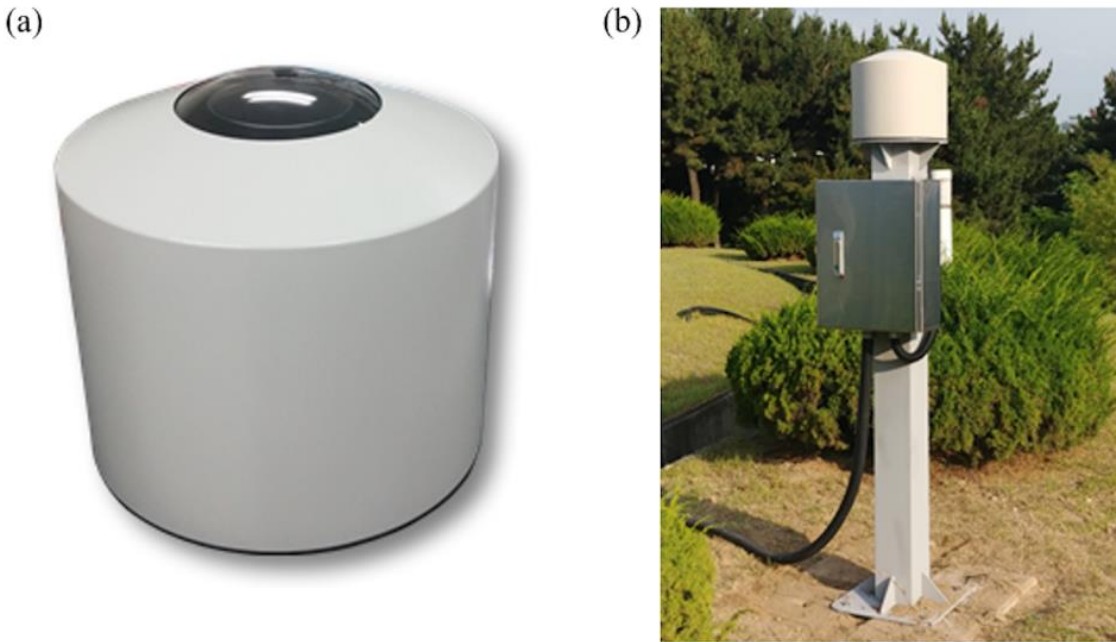


**Figure 1: Appearance of automatic cloud observation system (ACOS) (a) and installation environment (b) (Kim and Cha, 2020).**

**Table 1: Detailed automatic cloud observation system (ACOS) specifications.**

| Function | Description |
|---|---|
| Size | 264 mm (L) × 264 mm (W) × 250 mm (H), 6.5 kg |
| Pixels | 2432 × 2432 |
| Focal length | 8 mm, 180° fisheye lens |
| Sensor | CMOS |
| Aperture | F8 (daytime) ~ F11 (nighttime) |
| Sutter speeds | 1/1,000 s (daytime) ~ 5 s (nighttime) |
| ISO | 100 (daytime) ~ 25600 (nighttime) |
| Observation periods | 24 h operation, hourly observation for 10 min |
| Etc. | 24 h automatic heating (below –2°C) and ventilation |

The ACOS captures a sky-dome similar to human-eye observation and saves it as a two-dimensional image. However, images captured with a fisheye lens are distorted because the size of the objects placed at the edge of the image is relatively smaller than that at the center of the image (Lothon et al., 2019). Therefore, we performed an orthogonal projection distortion correction for the relative sizes of the objects in the image (Kim et al., 2021b). In addition, because the FOV of the imager is 180°, the horizontal plane is permanently shielded by surrounding objects (buildings, equipment, and trees) (Kim et al., 2015; Kim et

al., 2016). Therefore, only the pixel data within a FOV of 160° (zenith angle of 80°) were used in this study. In addition, the



image produced by the imager was converted into $128 \times 128$ pixels and used for training, validating, and testing the CNN model, thus ensuring the estimation of cloud cover even in a resource-constrained DL environment.

**2.2 Verification dataset**

Traditional cloud cover observation estimates the amount of cloud cover in the sky by observing visible clouds from the ground
(Spänkuch et al., 2022; WMO, 2021). In other words, by observing clouds in various directions of the sky-dome through the human eye, the cloud cover is determined as a tenth of 0 to 10 by comprehensively estimating the amount of covered clouds using human cognitive abilities and previously learned memories. Therefore, in this study, the accuracy of the cloud cover estimated by the CNN was evaluated by considering the human-eye observation (OBS) cloud cover as the true value. In addition, to determine the suitability of camera-based ground cloud cover estimation, its estimation performance was compared
with the cloud cover estimation performance of meteorological SAT and CEI. We used accuracy, bias, root mean square error (RMSE), and correlation coefficient (R) for a comparative analysis of the data using Eqs. (1)–(4).

$$\text{Accuracy} = \frac{TP + TN}{TP + TN + FP + FN} \tag{1}$$

$$\text{Bias} = \frac{\sum(M - O)}{N} \tag{2}$$

$$\text{RMSE} = \sqrt{\frac{\sum(M - O)^2}{N}} \tag{3}$$

$$\text{R} = \frac{\sum(M - \overline{M})(O - \overline{O})}{\sqrt{\sum(M - \overline{M})^2}\sqrt{\sum(O - \overline{O})^2}} \tag{4},$$

where *TP*, *TN*, *FP*, and *FN* denote the number of true positives, true negatives, false positives, and false negatives, respectively; *O* denotes the observed cloud cover; *M* denotes the estimated cloud cover (CNN, SAT, or CEI); and *N* denotes the number of data.


For the meteorological SAT, cloud cover data from GeoKOMPSAT-2A (GK-2A), a geostationary SAT of the KMA National Meteorological Satellite Center (NMSC), were used. The cloud cover of GK-2A was estimated using a cloud fraction within a radius of 5 km after converting the Cartesian coordinates of the grid (resolution 2 km × 2 km) around the reference grid point into spherical coordinates. In this case, considering the cloud height, zenith angle, and cloud cover observed on the ground, an
approximation of the cloud cover on the ground was determined using a regression equation to which weights under each condition were applied (NMSC, 2021). Because these data are provided as integer values from 0 to 100%, they were converted into tenths from 0 to 10, as listed in Table 2. The CEI (Vaisala CL31) used in this study uses cloud detection information sampled four times per minute to estimate the cloud cover by weighting the information for 30 min. As the estimated cloud cover was recorded in oktas, it was converted into tenths, as summarized in Table 2. In the case of 2 and 6 oktas, they were



converted into tenths (2 or 3 tenths and 7 or 8 tenths) with the smallest difference from the observed cloud cover at the same observation time (if the observed cloud cover was 3 tenths, the 2 oktas were converted into 3 tenths). The time resolution of the two datasets was determined at the same 1 h interval as the observation interval, and missing data were excluded from the test dataset.

**Table 2: Tenth cloud cover conversion table of satellite (%) and ceilometer (okta) cloud cover.**

| % | ≤5 | 5–15 | 15–25 | 25–35 | 35–45 | 45–55 | 55–65 | 65–75 | 75–85 | 85–95 | >95 |
|---|---|---|---|---|---|---|---|---|---|---|---|
| Okta | 0 | 1 | 2 | 2 | 3 | 4 | 5 | 6 | 6 | 7 | 8 |
| Tenth | 0 | 1 | 2 | 3 | 4 | 5 | 6 | 7 | 8 | 9 | 10 |

**3 CNN model architecture**

A CNN is a DL method used in various computer vision applications, including image classification and object detection. For tree-, vector-, and regularization-based ML methods, the models are trained using predefined features (Kim et al., 2022a; Kim et al., 2022b; Kim et al., 2022c). Therefore, when using an ML method, it is necessary to understand the chromatic statistical characteristics of the input data before constructing a model (Kim et al., 2021b). By contrast, DL methods, such as CNNs, extract spatial characteristics from the input image while iteratively performing forward and backward propagation, enabling the model to learn the features of the image (Ye et al., 2017). This process can enable the design of a highly accurate model by setting hyperparameters such as layer depth, feature map size, activation function, and learning rate (Wang et al., 2020). In this study, the CNN model comprised an input layer, seven convolutional layers (Conv), six pooling layers, three fully connected layers (Fc), and an output layer as shown in Fig. 2, and the hyperparameters of each layer were set as follows.

The image is inputted into the input layer as $128 \times 128 \times 3$ three-dimensional tensor data comprising $128 \times 128$ RGB channels. At each step of the convolutional layer, several $n \times n$ filters (kernels) scan the input data and extract their convolved feature maps. The filter of the convolutional layer has a weight associated with a specific area of the image and recognizes a specific pattern or structure of the image by learning the weight (Yao et al., 2021). In this study, a $3 \times 3$ filter was used, and zero padding was used to maintain the feature map characteristics of multiples of two. At each stage of the pooling layer, each feature map is downsampled to a size of $1/n$ using an $n \times n$ size filter to reduce the image size. The pooling process is used to abstract images and improve the generalizability of the model (Zhou et al., 2021). In addition, this process avoids overfitting, and the prediction accuracy is improved because fewer unnecessary details are learned in addition to the main features. Max pooling, which extracts the maximum value within a $2 \times 2$ filter, is used (Geng et al., 2020). In the fully connected layer, the feature map output from the last convolutional layer is input and flattened to one dimension to estimate the cloud cover using a multilayer perceptron neural network. In this process, the first fully connected layer randomly reduces 10% of the neurons (dropout = 0.1) to avoid overfitting (Srivastava et al., 2014).



An activation function exists between the convolutional, pooling, and fully connected layers, which converts the input signal such that it has nonlinear characteristics before being transmitted to the output signal. Because of the nature of the CNN model, as the convolutional layer deepens, the problem of exploding or vanishing weights may occur. We update the weights using a leaky rectified linear unit (LeakyReLU = 0.1) activation function to address these two problems (Yuen et al., 2021). In the last fully connected layer, the probability distribution of the 11 cloud cover classes is obtained using the softmax activation function, and the class with the highest probability is classified as a cloud cover.

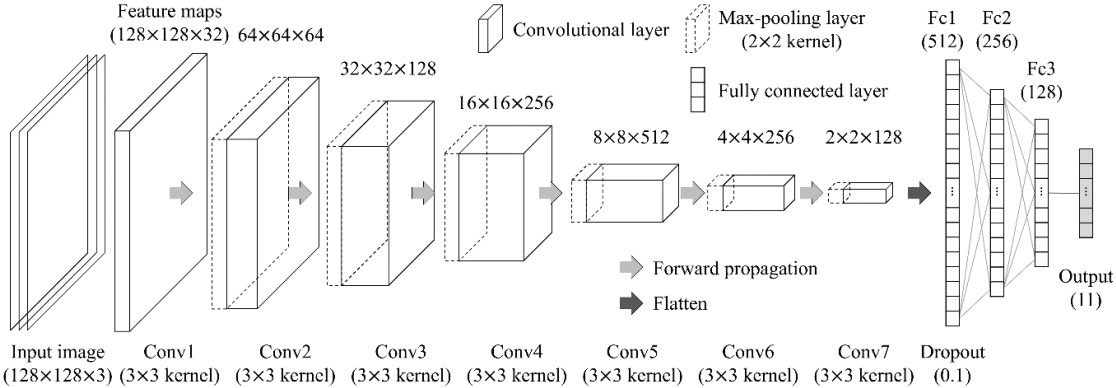

**Figure 2: Architecture of CNN model for cloud cover classification.**

In the output layer, the error between the output value and correct answer (label) is minimized using gradient descent while adjusting the weights of each layer, whereas forward and backward propagation is repeated. We used adaptive moment estimation (ADAM) for the gradient descent. ADAM is a combination of momentum optimization and root mean square propagation algorithms and is an optimization algorithm with excellent performance (Onishi and Sugiyama, 2017). The learning rate of the CNN model was set to 0.001, and the number of data points used for learning once per epoch (batch size) was set to 256. The training and validation results of the CNN model are evaluated in terms of categorical loss (mean square error [MSE]) and accuracy by epoch, as shown in Fig. 3. If the validation loss of the learned result did not improve compared to the loss before the fifth epoch, the weight of the epoch with the lowest previous loss was selected to determine the optimal CNN model. Consequently, the estimation performance of the model with the weights updated 70 times (70 epochs) was the best, that is, compared with the OBS cloud cover, the model achieved a bias, RMSE, and R of –0.04 tenths, 0.67 tenths, and 0.98 on the training dataset, and –0.03 tenths, 1.00 tenths, and 0.96 on the validation dataset, respectively. This result is an improvement compared to the cloud cover estimation performance on the training (bias, RMSE, and R of 0.07 tenths, 1.05 tenths, and 0.96, respectively) and validation (bias, RMSE, and R of 0.06 tenths, 1.51 tenths, and 0.93, respectively) datasets achieved by the supervised learning-based support vector regression method presented by Kim et al. (2021b).



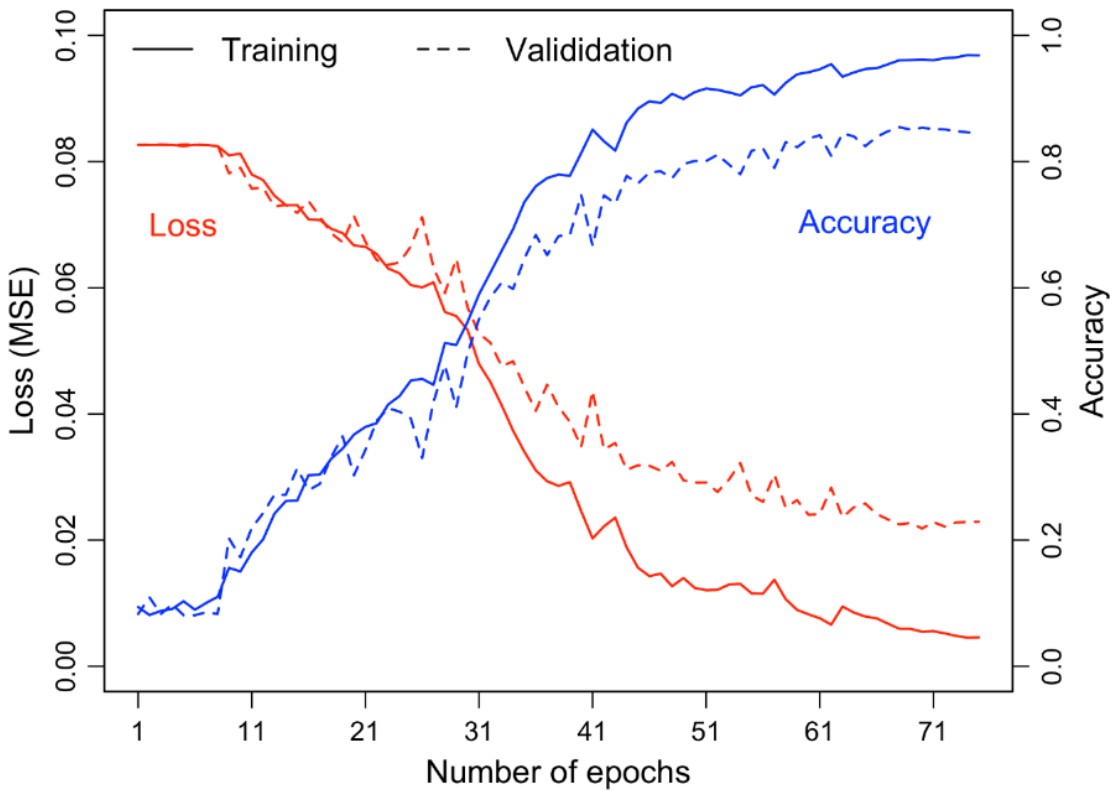

**Figure 3: Categorical loss and accuracy by epoch for training and validation datasets.**

## 4 Results

### 4.1 Evaluation of the CNN model on the test dataset

The CNN and OBS cloud covers estimated using the test dataset are shown as density heatmap plots for all cases and seasonal cases in Fig. 4. The column in each plot indicates the ratio (%) of the cloud cover estimated by the CNN to those by OBS, that is, a higher frequency in the diagonal one-to-one grids results in a higher agreement between the OBS and CNN cloud covers.

The Korean Peninsula shows various distributions and visually different characteristics of cloud cover owing to the influence of seasonal air masses and geographical characteristics (Kim et al., 2021b). In other words, in winter, the sky is generally clear, and cloud occurrence frequency and cloud height are low owing to the influence of the Siberian air mass; in summer, the weather is generally cloudy, and cloud occurrence frequency and cloud height are high owing to the influence of the Okhotsk Sea and North Pacific air mass; and in spring and autumn, the weather is fluid owing to the influence of the Yangtze River air

mass (Kim and Lee, 2018; Kim et al., 2020a; Kim et al., 2021a). In addition, the Korean Peninsula is located in the westerly wind zone, and cumulus heat clouds generated in the West Sea flow inland and develop (Kim et al., 2020b). The distribution of cloud cover by season is shown in Figs. 4b–4e, and the test results of the estimated cloud cover are summarized in Table 3.





By season, the cloud cover of 0 and 10 tenths had a high agreement, and the spread between 1 and 9 tenths was large; however, it generally exhibited a linear distribution with that of OBS. The CNN cloud cover exhibited a small difference from that of

OBS in terms of seasonal mean, with an accuracy of 0.93 or higher and a high correlation coefficient of 0.91 or higher. The evaluation of the CNN cloud cover for all cases exhibited an accuracy, RMSE, and R of 0.92, 1.40 tenths, and 0.95, respectively, indicating improved estimation performance compared to those described by Kim et al. (2021b) using an ML method (accuracy, RMSE, and R of 0.88, 1.45 tenths, and 0.93, respectively). Figure 5 shows the daily mean cloud cover of OBS and CNN for the test dataset. The daily mean estimation results also exhibited a bias, RMSE, and R of –0.15 tenths, 0.63 tenths, and 0.99,

respectively, indicating improved results compared to those described by Kim et al. (2021b) (RMSE and R of 0.92 tenths and 0.96, respectively).





**Figure 4: Density heatmap plots of observed (OBS) and estimated (CNN) cloud cover for all cases (a) and seasonal cases (b)–(e) for**
**the test dataset. The number of parentheses in each column denotes the number of OBS cloud cover cases.**





**Table 3: Seasonal accuracy, bias, RMSE, and R of estimated (CNN) cloud cover for the test dataset.**

| Season | N | Accuracy | Bias | RMSE | R |
|---|---|---|---|---|---|
| Winter | 1029 | 0.94 | −0.26 | 1.38 | 0.95 |
| Spring | 1448 | 0.94 | −0.02 | 1.53 | 0.93 |
| Summer | 1186 | 0.93 | −0.13 | 1.31 | 0.91 |
| Fall | 1079 | 0.93 | −0.16 | 1.33 | 0.95 |
| All season | 4742 | 0.92 | −0.13 | 1.40 | 0.95 |

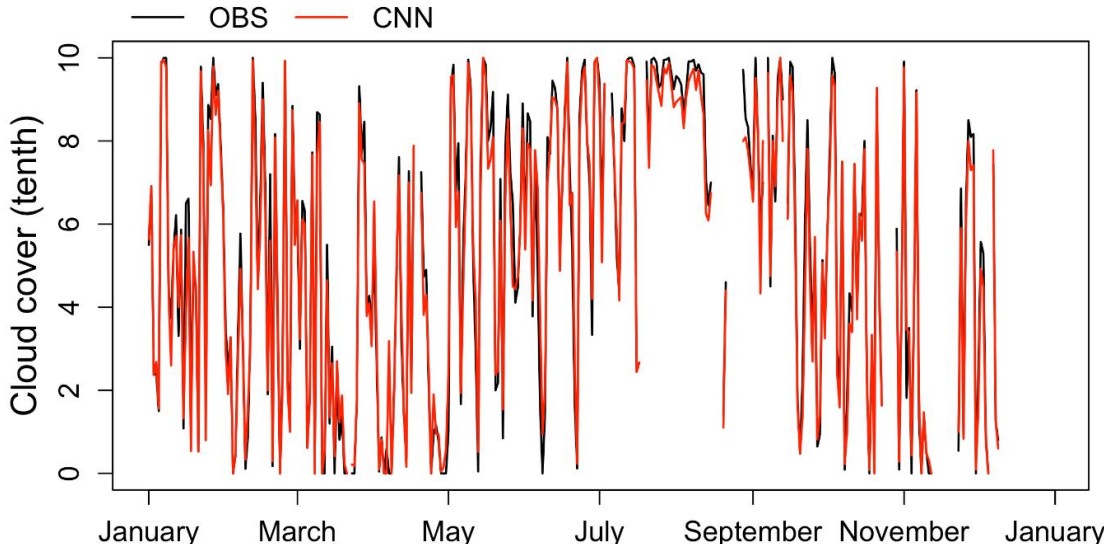

**Figure 5: Observed (OBS) and estimated (CNN) daily mean cloud cover time series for the test dataset.**

The CNN cloud cover during daytime, nighttime, and sunrise/sunset time is summarized in Table 4. In this study, daytime was defined as a solar zenith angle (SZA) of less than 80°, nighttime was defined as an SZA > 100°, and sunrise/sunset time was defined as $100° \geq SZA > 80°$. In general, the daytime and nighttime CNN cloud cover did not exhibit a large difference

compared with the OBS cloud cover; however, the RMSE was relatively large, and R was low during sunrise/sunset time. This is because the sky and clouds become reddish or bluish owing to the skyglow during sunrise/sunset time, making it difficult to distinguish between the sky and clouds (Kim and Cha, 2020; Kim et al., 2016; Kim et al., 2021b). Humans observe the sky-dome in three dimensions and easily detect covered clouds within the skyglow based on previous observations; however, there are limitations in the method using only limited information (images) such as this study (Al-lahham et al., 2020; Krinitskiy et

al., 2021). Moreover, unsupervised learning-based DL methods (e.g., segmentation and clustering) can generate large errors (Fa et al., 2019; Xie et al., 2020). These methods have the advantage of being able to estimate cloud cover without learning



previously accumulated data. However, because there is no correct answer, the estimation performance deteriorates if the sky and clouds are not clearly distinguished, as in these limitations (Zhou et al., 2022). Therefore, optical image correction for a sky-dome such as that described by Hasenbalg et al. (2020) will be required to estimate the cloud cover from these images.

Nevertheless, this study achieved better cloud cover estimation results than those of Kim et al. (2016) for daytime (RMSE and R of 2.12 tenths and 0.87, respectively) and those of Kim and Cha (2020) for nighttime (RMSE and R of 1.78 tenths and 0.91, respectively) using the threshold method. In addition, the accuracy and R by time exhibited improved results than those of Kim et al. (2021b) (accuracy and R values of 0.89 and 0.95, 0.86 and 0.93, and 0.85 and 0.90 for daytime, nighttime, and sunrise/sunset time, respectively) using the ML method.


**Table 4: Accuracy, bias, RMSE, and R for daytime, nighttime, and sunrise/sunset time of estimated cloud cover for the test dataset.**

| Time | N | Accuracy | Bias | RMSE | R |
|---|---|---|---|---|---|
| Daytime | 2163 | 0.93 | –0.12 | 1.27 | 0.95 |
| Nighttime | 1838 | 0.93 | –0.08 | 1.39 | 0.95 |
| Sunrise/sunset | 741 | 0.90 | –0.29 | 1.74 | 0.92 |

The relative frequency distribution by season and time of cloud cover difference between the OBS and CNN cloud cover is shown in Fig. 6. In this relative frequency distribution, a higher frequency at which the difference is 0 tenth results in a higher

agreement between the OBS and CNN cloud cover observations. In general, a comparison between automatic instrument observations and cloud cover observed by humans allows for a difference of two levels (i.e., 2 tenths or 2 oktas) (Ye et al., 2022). Table 5 summarizes the agreement for the 0–3 tenths cloud cover difference between the OBS and CNN. The agreement of the difference between 0 and 2 tenths was greater than approximately 61%, 83%, and 91% for all seasons, and the high agreements were 63.79%, 84.65%, and 92.66% for all cases, respectively. This result is an improvement of approximately

22%, 5%, and 3%, respectively, compared with the agreement for the 0–2 tenths difference reported by Kim et al. (2021b) using the ML method. During nighttime and sunrise/sunset time, the agreement for a 0-tenth difference between the OBS and CNN cloud cover improved significantly to approximately 26% and 27%, respectively, whereas that for a 1-tenth difference improved to approximately 8% and 11%, respectively. The CNN cloud cover in this study exhibited a high agreement of approximately 93% with that of OBS within a difference of 2 tenths and exhibited a higher agreement than 80–91% agreements

of previous studies using the threshold, ML, and DL methods (Fa et al., 2019; Kim and Cha, 2020; Kim et al., 2015; Kim et al., 2016; Kim et al., 2021b; Krinitskiy and Sinitsyn, 2016; Wang et al., 2021; Xie et al., 2020).





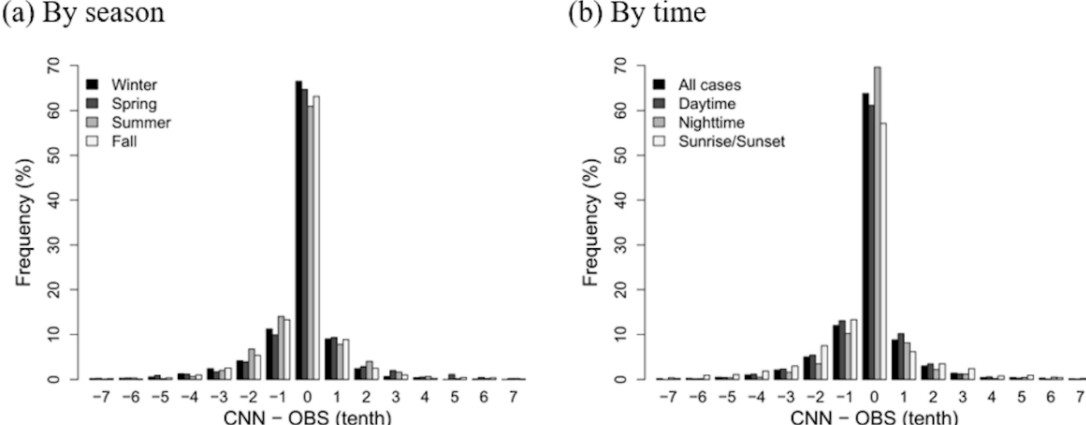

**Figure 6: Relative frequency distributions of differences between observed (OBS) and estimated (CNN) cloud cover by season and time for the test dataset.**

**Table 5: Agreements of differences (Diff.) between observed (OBS) and estimated (CNN) cloud cover by season and time for the test dataset.**

| Diff. | Winter | Spring | Summer | Fall | Annual | Daytime | Night-time | Sunrise/sunset |
|---|---|---|---|---|---|---|---|---|
| ±0 tenth | 66.47 | 64.71 | 60.96 | 63.11 | 63.79 | 61.12 | 69.64 | 57.09 |
| ±1 tenth | 86.78 | 84.05 | 82.88 | 85.36 | 84.65 | 84.47 | 88.08 | 76.65 |
| ±2 tenths | 93.39 | 90.88 | 93.68 | 93.23 | 92.66 | 93.39 | 93.80 | 87.72 |
| ±3 tenths | 96.50 | 94.61 | 97.30 | 96.85 | 96.20 | 96.99 | 96.52 | 93.12 |

**4.2 Verification with satellite and ceilometer data**

To determine the suitability of the cloud cover estimation method using the camera-based imager presented in this study, OBS, SAT, and CEI cloud cover data were compared. For comparison, cloud cover data from 4,634 cases were used, excluding data with missing SAT or CEI cloud cover in the test dataset. A density heatmap plot of the OBS cloud cover and the SAT and CEI cloud covers is shown in Fig. 7. Unlike the density heatmap plot of the CNN cloud cover, the SAT and CEI cloud covers showed overestimation or underestimation of the cloud cover. In other words, the frequencies of OBS and CNN cloud cover

were extremely similar in the relative frequency distribution by cloud cover, as shown in Fig. 8, whereas SAT cloud cover had a high frequency in 10 tenths and a low-frequency distribution in other mostly cloudy cases. Conversely, the CEI cloud cover





exhibited a low-frequency distribution in the 10 tenths and a high-frequency distribution in partly cloudy cases. The SAT and CEI cloud cover evaluation results are summarized in Table 6. Both remote observation results exhibited low accuracy, large bias, low RMSE, and low R values and agreements. Although SAT data have several spatial and temporal advantages, large

cloud detection errors occur due to the large spatial resolution of 2 km × 2 km and uncertainty in cloud cover estimation based on cloud height (NMSC, 2021; Zhang et al., 2018). Using a CEI, it is difficult to accurately detect and estimate clouds located in the sky-dome by observing only a narrow beam-width area (Utrillas et al., 2022). Therefore, to estimate cloud cover from the ground, the combination of images acquired with a camera-based imager and a CNN is the most suitable and closest method to replace human-eye observation.


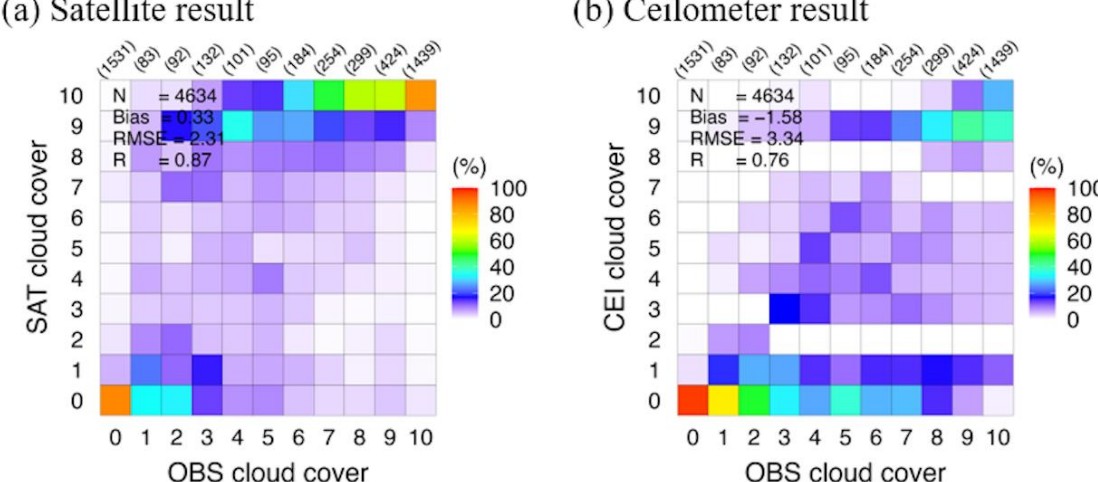

**Figure 7: Density heatmap plots of satellite (SAT) and ceilometer (CEI) cloud covers for the test dataset. The number of parentheses in each column denotes the number of OBS cloud cover cases.**



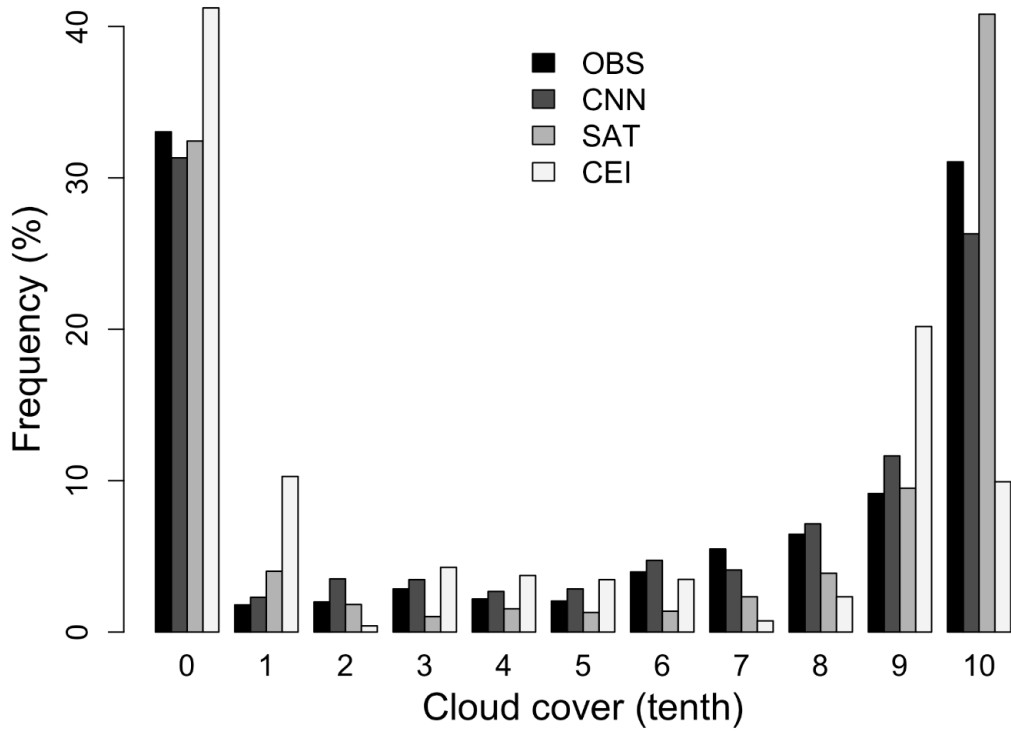

**Figure 8: Relative frequency distribution by cloud cover of observation (OBS), estimation (CNN), satellite (SAT), and ceilometer (CEI) cloud cover for the test dataset.**

**Table 6: Accuracy, bias, RMSE, R, and agreements of satellite and ceilometer for the test dataset. The unit of bias and RMSE is tenths, and agreement for the 0–3 tenths differences is %.**

|      | Accuracy | Bias  | RMSE | R    | ±0 tenth | ±1 tenth | ±2 tenths | ±3 tenths |
|------|----------|-------|------|------|----------|----------|-----------|-----------|
| SAT  | 0.89     | 0.33  | 2.31 | 0.87 | 59.09    | 73.87    | 82.82     | 88.78     |
| CEI  | 0.86     | −1.58 | 3.34 | 0.76 | 46.72    | 66.81    | 73.54     | 77.13     |

## 5 Summary and conclusions

In this study, images captured using a camera-based imager and a CNN were used to estimate 24 h continuous cloud cover from the ground. Data collected over a long period were used to capture various visual clouds and estimate cloud cover. Images were captured by a manned weather station from 2019 to 2020 at 1 h intervals, matching the time interval typically used for human-eye observations. The 2019 data were used for training and validating the CNN model, whereas the 2020 data were used for testing. The training dataset did not use the entirety of the collected data but used randomly sampled data with replacements for each cloud cover class to organize the dataset. In other words, overfitting of the cloud cover class with a high



observation frequency was prevented, and weight was assigned to the class with a low observation frequency. The estimated results were compared with observational data from a manned weather station and other remote observational data.

Consequently, the cloud cover estimated for the test dataset exhibited an accuracy, RMSE, R, and agreement of 0.92, 1.40 tenths, 0.95, and within ±2 tenths difference of approximately 93%, respectively, with OBS cloud cover. This result shows improved cloud cover estimation performance compared with those of previous studies using the threshold, ML, and DL methods. In addition, the camera-based imager with a CNN was found to be the most suitable for cloud cover estimation on the ground compared to the estimation using a SAT and CEI. SAT and CEI remote observations can determine the temporal,

spatial, and vertical distributions of clouds; however, their uncertainty is extremely large. A camera-based imager with a CNN, as in this study, is the most suitable method for replacing ground cloud cover observations. This configuration can be fully utilized, even in a limited computer resource environment, using a low-cost fisheye camera-based imager and edge computing. The formation of these dense observation networks and the accumulation of data make it possible to maintain the consistency of meteorological data. Therefore, various observation devices and methods that can replace cloud observation methods that

use human-eye observations on the ground should be developed and tested.

**Code availability:** The code for this paper is available from the corresponding author.

**Sample availability:** The sample for this paper are available from the corresponding author.

**Author contribution:** BYK carried out this study and the analysis. The results were discussed with JWC and YHL. BYK

developed the deep learning model code and performed the simulations and visualizations. The manuscript was mainly written by BYK with contributions by JWC and YHL.

**Competing interests:** The authors declare that they have no conflict of interest.

**Acknowledgments:** This work was funded by the Korea Meteorological Administration Research and Development Program "Research on Weather Modification and Cloud Physics" under Grant (KMA2018-00224).

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
