# Peer review of "Estimation of twenty-four-hour continuous cloud cover using groundbased imager with convolutional neural network"

_Atmospheric Measurement Techniques, 2023_

## Author Response (AR2)

**Reviewer #1**

General Comments:

Accurate cloud cover observations are crucial for analyzing the macro and micro characteristics of clouds and studying their impact on weather and climate. This study utilizes a camera-based imager and convolutional neural network (CNN) as a prospective alternative to ground-based human-eye observations. The optimized CNN model employed in this research yields favorable performance metrics, exhibiting relatively smaller errors compared to traditional alternatives, thereby demonstrating certain advantages and holding scientific value. However, in comparison to other studies that employ machine learning algorithms for cloud cover analysis (including the author's previous works), the improvements presented in this paper are relatively limited, and the level of innovation is less prominent. Therefore, it is recommended to reconsider the acceptance of this study after the following issues have been solved.

**Thank you for reviewing this manuscript. Based on the reviewer's comments, we have added some content to the revised manuscript, as detailed below. We believe that the quality of the manuscript has been improved and the clarity of the manuscript has been enhanced after addressing the reviewer's comments.**

Specific Comments:

1) Compared to the study introduced in Kim et al. (2021b), the improvement in cloud cover analysis achieved by the algorithm presented in this study appears to be limited. Please provide a detailed explanation of the novelty and necessity of this study.

**We have added related contents as follows:**

**L179–192 "Clouds exhibit varying colors at different times of the day, including daytime, nighttime, and sunrise/sunset time; therefore, the threshold method (traditional method), which uses the ratio or difference of RGB brightness, cannot effectively distinguish the clouds from the sky. Using the threshold method, Kim et al. (2016), Shields et al. (2019), and Kim and Cha (2020) estimated the cloud cover by dividing the daytime and nighttime algorithms. In this case, estimated cloud cover at sunrise and sunset time may appear discontinuous, and the degree of uncertainty is large. Furthermore, depending on the shape, thickness, and height, the clouds in the image can appear dark, bright, or transparent. Therefore, it is necessary to distinguish the sky and clouds using the spatial characteristics of the image. However, Kim et al. (2021) discovered that an ML method that learns using the statistical characteristics (mean, kurtosis, skewness, quantile, etc.) of the RGB color in the image can reflect the nonlinear visual characteristics effectively, but not the spatial characteristics. Therefore, a DL method capable of accurately reflecting the visual and spatial characteristics of an image, such as the one utilized in this study, is suitable for estimating cloud cover for a sky-dome. In this study, more data were used for training each cloud cover class through random sampling with a replacement than those used by Xie et al. (2020) and Ye et al. (2022). Furthermore, using DL supervised learning, the ability to extract image features was further improved by using deeper convolutional layers compared with Onishi and Sugiyama (2017)."**

**L310–314 "In this study, a novel method was attempted to learn a DL model for cloud cover estimation. Compared to the datasets of previous studies (e.g., Fa et al., 2019; Kim et al., 2021b;**

**Xie et al., 2020; Ye et al., 2022), more images were learned, and long-term estimated data were analyzed. Furthermore, the estimated results were compared with observational data from a real weather station and other remote observational data (i.e., from a satellite and ceilometer)."**

**L241–247 "Moreover, in this study, images acquired at sunrise/sunset time accounted for 16.23% of all training datasets. In other words, the images acquired at sunrise/sunset time were trained 2 to 3 times lower images for each cloud cover class than images acquired at daytime and nighttime. The DL method can degrade training performance when the amount of labeled data is limited (Ker et al., 2017). Conversely, the DL method can extract image features with a more complex structure by more complex and deeper learning as the amount of data increases (LeCun et al., 2015). Therefore, it is expected that more robust and accurate results can be obtained if more images are acquired during sunrise/sunset time (Geng et al., 2021; Qian et al., 2022)."**

2) Figure 5&6: From these figures, it can be observed that the algorithm tends to underestimate cloud cover when compared to the observed results. Please provide an explanation for this phenomenon.

**We have revised the manuscript by adding the following sentences:**

**L227–229 "In general, the daytime and nighttime CNN cloud cover did not exhibit a large difference compared with the OBS cloud cover; however, the bias and RMSE were relatively large, and R was low during sunrise/sunset time."**

**L233–236 "In particular, there were many cases where the CNN cloud cover was smaller than the OBS cloud cover in the 9 and 10 tenths classes during sunrise/sunset time. The mean SZAs of the two classes were 85.13° and 98.11°, respectively, and the error was relatively large when the sun moved completely above and below the horizon (i.e., at 7–8 LST and 19–20 LST, respectively)."**

3) P2L4: satellites can't be categorized into "ground-based methods".

**We have revised existing text as follows:**

**L33–34 "Remote and automatic observation as well as estimation of the cloud cover on the ground can be achieved using meteorological satellites (SATs), ceilometers (CEIs), and camera-based imagers."**

**Reviewer #2**

This paper proposes a novel method to estimate ground-level cloud cover using images from a ground-based sky imager and convolutional neural network, as an alternative to human observations. The topic is of great value and the methodology is generally sound. The paper is well structured and well written. The study provides a feasible technical approach for automated monitoring of surface cloud cover. In summary, the following issues need further improvement:

**Thank you for reviewing this manuscript. Based on the reviewer's comments, we have added some content to the revised manuscript, as detailed below. We believe that the quality of the manuscript has been improved and the clarity of the manuscript has been enhanced after addressing the reviewer's comments.**

(1)The input data of photos were sampled at 1 hour intervals, corresponding to the manual observations. It is unclear whether the sampling time is exactly the same as the time of manual observation. Clouds can change rapidly in a short period of time, which may lead to inaccurate sample data if there is a time discrepancy between the photos and observations.

**We agree with the reviewer's comments. The KMA surface weather observation guidelines stipulate that observations should be conducted for every hour on time. However, the timing of observations can differ depending on the observer, which can lead to inconsistencies and potential inaccuracies in the collected data. Unfortunately, such information has not been documented, preventing us from determining the precise observation time. Therefore, we have revised and added the following sentence:**

**L67–68 "The image data used were captured at 1 h intervals and on time from January 2, 2019, to December 9, 2020, considering the human-eye observation interval."**

**L68–69 "According to the KMA surface weather observation guidelines, cloud cover should be observed every hour on time (KMA, 2022). However, a slight difference in observation time may occur, depending on the observer. In this study, it was assumed that the time difference between the images observed on time and the human observations would not be significant."**

(2)This study utilizes CNN to establish the model, with images as input and manual observations as true value labels. The method is relatively effective. However, there are two aspects of problems:

1) Manual observations also have errors, which are determined by human subjective judgments. Therefore, the methods developed based on such samples and labels inevitably still contain errors, which cannot be further improved through model tuning and data augmentation. It is recommended to try comparison with other conventional methods, and develop research methods to improve the accuracy of sample data;

2) The manual observations quantify the results into 0-10 tenths. From the observation perspective, the resolution of such cloud cover observations is slightly low. Using 1 tenth to 1.9 tenths may have a difference of 9 percentage points, which needs further improvement in the observation accuracy of cloud cover for refined observations and forecasts.

We agree with the reviewer's comments on 1) and 2). The KMA-observed cloud cover was recorded as 0-8 oktas or 0-10 tenths according to the WMO synoptic observation guidelines. The employed observation method, as also noted by the reviewer, has a low resolution of quantitative values; hence, the margin of error is a few % in the same cloud cover class. Therefore, we believe that it would be possible to estimate the cloud cover with high accuracy if the pixel information (with or without clouds) in the image is extracted and learned in terms of the percent cloud cover as the label. In this study, the data resolution was rather low because the OBS cloud cover was used as a label. Therefore, we have added the following sentence:

L321–323 "Depending on the characteristics of the data to be learned, it is possible to estimate the cloud cover in % instead of oktas and tenths; accordingly, the time resolution can also be estimated in minutes rather than hourly intervals."

(3)Based on traditional image processing methods like NBR threshold, the accuracy is usually lower during sunrise and sunset due to the influence of scattering near the sun. This study also found larger errors during this period. What are the reasons? Is it because there are fewer manual observation samples?

We used the DL method to compensate for the shortcomings of the threshold method and the ML method. The estimation accuracy of the cloud cover in this study was better than that of the previously employed threshold method and ML method. However, the accuracy was still relatively low at sunrise/sunset time. This is because, as noted by the reviewer, the image was polluted due to the scattering of light near the sun, making it difficult to distinguish clouds from the sky. Therefore, we added the following sentence to the main text on possible methods to improve the accuracy of cloud cover estimation at sunrise/sunset time:

L241–247 "Moreover, in this study, images acquired at sunrise/sunset time accounted for 16.23% of all training datasets. In other words, the images acquired at sunrise/sunset time were trained 2 to 3 times lower images for each cloud cover class than images acquired at daytime and nighttime. The DL method can degrade training performance when the amount of labeled data is limited (Ker et al., 2017). Conversely, the DL method can extract image features with a more complex structure by more complex and deeper learning as the amount of data increases (LeCun et al., 2015). Therefore, it is expected that more robust and accurate results can be obtained if more images are acquired during sunrise/sunset time (Geng et al., 2021; Qian et al., 2022)."